# Creation of a Biobank of the Sperm of the Honey Bee Drones of Different Subspecies of *Apis mellifera* L.

**DOI:** 10.3390/ani13233684

**Published:** 2023-11-28

**Authors:** Alexey N. Gulov, Andrey S. Berezin, Elena O. Larkina, Elena S. Saltykova, Milyausha D. Kaskinova

**Affiliations:** 1Federal State Budgetary Scientific Institution «Federal Beekeeping Research Centre», Pochtovaya Street, 22, Ryazan Region, 391110 Rybnoe, Russia; mellifera@yandex.ru (A.S.B.); elena.72larkina@gmail.com (E.O.L.); 2Institute of Biochemistry and Genetics—Subdivision of the Ufa Federal Research Centre of the Russian Academy of Sciences, Prospekt Oktyabrya 71, 450054 Ufa, Russia; esaltykova1960@gmail.com

**Keywords:** *Apis mellifera*, honey bee subspecies, cryopreservation, breeding stock, honey bee selection, *tRNAleu-COII (COI-COII)*, microsatellite markers, morphometry

## Abstract

**Simple Summary:**

This study provides an example of the creation of a biobank of drone sperm of specific subspecies of *Apis mellifera* L. For subspecies identification, morphometric (“classical morphometry”) and genetic (*tRNAleu-COII* mtDNA and nine SSR markers) methods were used. A honey solution was used as a cryopreservative. It was shown that the main values of frozen–thawed sperm decreased compared to those of fresh sperm. However, the use of honey in sperm cryopreservation has great potential.

**Abstract:**

The cryopreservation of gametes and embryos is an important element of biodiversity conservation. One species in need of conservation is the honey bee *Apis mellifera* L. Changing environmental factors, especially the anthropogenic factor, have led to a reduction in the numbers of this insect species. In this study, we provide an example of the creation of a biobank of honey bee drone sperm. For sperm cryopreservation, drones of the most common subspecies of honey bees common in Russia were selected. These were the dark forest bee, *Apis mellifera mellifera*, from the Republic of Bashkortostan, with three subspecies (*A. m. carnica*, *A. m. carpatica*, and *A. m. caucasica*) from the southern regions of Russia, as well as two breeding stocks, the Far Eastern bee and Prioksky bee. For subspecies identification, morphometric and genetic methods were used. The subspecies of the studied samples were confirmed via the analysis of the *tRNAleu-COII* locus of mitochondrial DNA and nine microsatellite markers of nuclear DNA. It was shown that bees of the Prioksky breeding stock belong to the subspecies *A. m. caucasica* based on phylogenetic analysis, and the Far Eastern breeding stock is a stable hybrid, descending on the maternal line from the evolutionary lineage C or O. The results of the morphometric analysis are consistent with the results of the genetic analysis. For the cryopreservation of sperm, we used a cryoprotectant solution with honey. As a result, the viability of frozen–thawed sperm decreased by 20.3% compared to fresh sperm, and overall motility decreased 25-fold. The measurement of the sperm concentration in the spermatheca of artificially inseminated queens showed that it varied from 0.22 to 4.4 million/μL. Therefore, the use of honey in sperm cryopreservation has great potential.

## 1. Introduction

The cryopreservation of gametes and embryos is an important tool for preserving the biodiversity of both wild and farmed animals and plants [1,2,3,4]. Research related to the possibility of the long-term preservation of honey bee sperm in liquid nitrogen was actively carried out around the world at the end of the last century. One of the founders of the cryopreservation of drone sperm [5], who used 25% dimethyl sulfoxide (DMSO) as a cryoprotectant, obtained 8% bee brood from queens artificially inseminated with frozen–thawed sperm. The follow-up results on the artificial insemination of queens with frozen–thawed sperm were published several decades later. Hopkins et al. (2012) [6] pioneered the use of egg yolk as a cryoprotectant for drone sperm. According to their data, out of five inseminated queens, only two produced offspring with more than 50% worker bees. The remaining queens had mixed broods or only drone broods. In order to reduce the toxic effects of DMSO on the quality characteristics of sperm, researchers used a citrate–HEPES buffer containing trehalose [7], soy lecithin [8], the seminal plasma of sheep, and honey bee drone sperm [9], with royal jelly as an extender [10]. The first successful experiments on the cryopreservation of drone sperm in the haemolymph of a honey bee were carried out in the USSR [11]. And, only at the beginning of this century, was a patent issued (No. 2173045 dated 10 September 2001; http://allpatents.ru/patent/2173045.html?ysclid=lmh7tuqzuy819222269, accessed on 1 September 2023) for the technology of the cryopreservation of drone sperm with the production of fertile queens inseminated with frozen–thawed sperm [12]. The cryopreservation of drone sperm was carried out on the basis of the C46 nutrient medium [12]. An assessment was made of the egg production of queen bees inseminated with frozen–thawed sperm after 25 years of storage in liquid nitrogen [13]. It was shown that sperm viability is maintained at a fairly high level (93%) and does not depend on the duration of cryopreservation. However, the fertilizing ability of these doses of sperm was low; the amount of capped worker brood from queens inseminated with frozen–thawed sperm was less than 50%. But research results have demonstrated the possibility of preserving drone sperm in liquid nitrogen for 25 years [13]. During long-term storage in liquid nitrogen, the sperm is exposed to a number of factors that cause structural and functional changes in cells. Damage to the acrosome and changes in chromatin condensation, induced by the processes of the freezing and thawing of the gamete, are clearly reflected in the morphology of sperm and its morphometric parameters [14]. The morphological parameters of sperm correlate more closely with the rate of fertilization than sperm concentration and motility [15,16]. Today, specialists from the Federal Beekeeping Research Centre (Russia, Ryazan region, Rybnoe) are developing a cryoprotectant solution with honey for the cryopreservation of drone sperm [17].

During the natural spread of the honey *bee Apis mellifera* L., it formed various subspecies and ecotypes adapted to specific climatic zones [18,19]. In addition to naturally occurring subspecies, work is being carried out to develop various breeding stocks and lines that differ in terms of their economically useful traits [20,21,22,23]. Due to human economic activity, the boundaries between subspecies began to blur, and some subspecies were absorbed by others [24,25]. To differentiate subspecies, morphometric and genetic methods are used. “Classical morphometry” includes the analysis of 36 morphometric features [18,26]. But, usually, no more than 10 features are used [27]. One of the most used genetic methods is the analysis of the polymorphism of the mitochondrial locus *tRNAleu-COII* (or *COI-COII*) [28,29]. Using the analysis of this locus, it is possible to differentiate honey bees from the evolutionary lineages A, M, C, O, and Y [18,19]. Allelic variants P(Q)_1−n_ are markers of the origin of bees from *A. m. mellifera* and *A. m. iberiensis* (lineage M), with allelic variant Q from subspecies from the evolutionary lineages C (*A. m. carnica*, *A. m. ligustica*) and O (*A. m. caucasica*, *A. m. anatoliaca*, *A. m. remipes*, *A. m. macedonica*) on the maternal line. The letters P and Q indicate repeats located between the *tRNAleu* and *COII* genes. Subspecies from the evolutionary lineages C and O lack the P repeat [29]. This mitochondrial marker allows us to establish the origin of bees only on the maternal line. To assess the drone background, microsatellite [30] and SNP [31,32,33] markers are used.

Consequently, beekeeping has the important role of preserving naturally occurring gene pools and gene pools obtained through selection for economically useful traits. And one of the tools for preserving the genetic diversity of honey bees is the cryopreservation of genetic material.

The aim of this study is to create a biobank of the sperm of *Apis mellifera* L. drones with a known origin on the basis of the Federal Beekeeping Research Centre (Rybnoye, Ryazan region, Russia, https://beecentr.ru/, accessed on 1 September 2023). To assess the origin of bees, morphometric (length of the proboscis; the length and width of the forewing and third tergite, tarsal index, and cubital index; the length of the third sternite; the length and width of the wax mirror; the distance between wax mirrors) and genetic (the analysis of the polymorphism of the microsatellite loci of nuclear DNA and the intergenic locus *tRNAleu-COII* of mitochondrial DNA) methods were used.

## 2. Materials and Methods

### 2.1. Sampling

Bees (drones and worker bees) were selected from 76 colonies (Table 1) belonging to four naturally occurring honey bee subspecies (*A. m. caucasica*, *A. m. carnica*, *A. m. carpatica*, and *A. m. mellifera*) and two breeding stocks (Far Eastern and Prioksky bees). Samples of the subspecies *A. m. mellifera* were selected from two districts of the Republic of Bashkortostan: from the Iglinsky district, in an apiary specializing in breeding *A. m. mellifera*, and from the Burzyansky district, an ancient reserve of the Burzyan population of *A. m. mellifera* [34]. The Grey Caucasian Mountain bee (*A. m. caucasica*) was collected at the Krasnopolyansk experimental beekeeping station in the Krasnodar Krai [35]. Samples of *A. m. carnica* and *A. m. carpatica* were selected in the Republic of Adygea. The Far Eastern bee was selected in the Primorsky Krai of the Far East, and Prioksky bees in the Ryazan region were obtained from the apiary of the Federal Beekeeping Research Centre.

### 2.2. Preparation of Honey Bee Drone Sperm

Sperm was collected from sexually mature drones at the age of 20–30 days by artificially stimulating endophallus eversion using SCHLEY-System model 1.04 equipment (A&G Wachholz, Espelkamp, Germany). One hundred microliters of sperm was collected from 110–125 drones from each bee colony. Freshly collected sperm samples were transported to the deposit site in glass capillaries (L = 90 ± 1.0 mm; d = 1.8 ± 0.2 mm) with a volume of 50 μL without the use of antibacterial contamination agents in a foam container with refrigerants within a temperature range of 2–8 °C.

Sperm quality was assessed by motility and membrane integrity by fluorescence microscopy using SYBR-14 and PI fluorochromes [36]. With the use of equipment for the instrumental insemination of queen bees, sperm with a volume of 50 μL was subjected to short-term storage at a temperature of 3 °C. After 2–3 months of storage in the refrigerator at 3 °C, the cryopreservation of the prepared sperm samples was performed. The extender included the following components: 10% honey (50 mL), lactose (10 mg), sucrose (10 mg), egg yolk (2.5 mL), and DMSO (5 mL) (10% of the volume of the honey solution). To prepare a 10% solution, honey was preheated in a water bath at 40–45 °C for 30 min. The concentration of hydrogen ions in the finished extender was adjusted with 6 M NaOH to a pH value of 8–9.

Next, to prepare one sample, 80 µL of freshly prepared extender and 10 µL of cooled sperm were added to a 1.5 mL Nunc cryovial (for 1 part sperm, 8 parts extender). All components were mixed until homogeneous and placed in a refrigerator for 1 h at 3 °C for equilibration.

The freezing of samples was carried out using a Bio Freeze BV-65 program freezer (Consarctic, Westerngrund, Germany). The protocol for freezing drone sperm at a rate of 3 °C/min was as follows:-start at 3 °C;-from 3 °C to −5 °C at a speed of 3 °C/min;-hold at −5 °C for 1 min;-from −5 to −12 °C at a speed of 1 °C/min;-hold at −12 °C for 9 min;-from −12 °C to −50 °C at a speed of 3 °C/min;-after −50 °C, drop the free temperature to −196 °C.

### 2.3. Instrumental Insemination of Queen Bees

The instrumental insemination of queen bees was carried out using SCHLEY-System model 1.04 equipment (A&G Wachholz). A single insemination was used with a volume of injected sperm of 10–12 μL. For insemination, virgin queens aged 7–8 days were used. The assessment of the reproductive values of artificially inseminated (AI) queens was replaced by an assessment of their physiological values—the concentration of sperm in the seminal receptacle and the presence of sperm residues in the paired oviducts [37].

The queens were dissected under an MBS-10 light microscope (Lytkarino Optical Glass Plant, Lytkarino, Russia). The presence or absence of sperm residues in the paired oviducts of the uterus was visually recorded. The seminal receptacle was freed from tissue and placed in a sterile 1.5 mL Eppendorf tube containing 250 μL of 10% honey extender. Then, the seminal receptacle was pierced with a needle, releasing the contents into the extender. After careful pipetting, a drop of the suspension was taken and Goryaev’s counting chamber was filled. Sperm counting and the determination of the sperm concentration were carried out according to a previously published method [13], taking into account that the volume of the seminal receptacle was 1 μL [38].

### 2.4. Morphometric Analysis

For morphometric analysis thirty worker bees from each colony were used. With the use of Altami Studio software version 3.5 (http://altamisoft.ru/, accessed on 1 September 2023), the following characteristics of bees were measured (Appendix A): proboscis length (*Lx*), fore wing length (*F_L_*) and fore wing width (*F_W_*), cubital index (*CI*, %), length of third tergite (*Lt3*) and width of third tergite (*Wt3*), length of third sternite (*Ls3*), length (*Lwm*) and width (*Wwm*) of the wax mirror, distance between wax mirrors (*Lwmd*), and the tarsal index *(TI*) [27].

Reference values are known for only three of the studied features: proboscis length (*Lx*), cubital index (*KI*, %), and the width of the third tergite (*Wt3*) [39,40]. Other values were compared between the studied samples.

### 2.5. Genetic Analysis

DNA was isolated from the thorax muscles of worker bees using the DNA-EXTRAN-2 kit (Syntol, LLC, Moscow, Russia). The quality and quantity of total DNA were analysed on an Implen N60 spectrophotometer (Implen GmbH, Munich, Germany).

To establish the maternal origin of bee colonies, an analysis of the mtDNA intergenic locus *tRNAleu-COII* was performed [28] using primers (5′-TCTATACCACGACGTTATTC-3′) and (5′-GATCAATATCATTGATGACC-3′). Subspecies from the evolutionary lineage M have allelic variants P(Q)_1−n_. Subspecies from the evolutionary lineages C (*A. m. carnica*, *A. m. carpatica*) and O (*A. m. caucasica*) have allelic variant Q.

To establish the drone background, an analysis of the polymorphism of nine microsatellite loci of nuclear DNA (*Ap243*, *4a110*, *A24*, *A8*, *A43*, *A113*, *A88*, *Ap049*, *A28*) was performed [30]. Samples of *A. m. mellifera* from the Burzyansky district and the Perm Krai (*N* = 136) were used as a reference group for the evolutionary lineage M. Samples from the Republic of Adygea, Krasnodar Krai, and Uzbekistan (*N* = 120) were used as representatives of the C and O evolutionary lineages.

PCRs were performed in a final volume of 20 μL:15 μL sterile deionised water, 2 μL of 10× PCR Buffer, 0.4 μL dNTP, 0.6 μL each primer (10 pmol/μL), 0.3 μL Taq DNA polymerase, and 2 μL DNA template. All PCR amplifications were carried out on a Bio-Rad T100 thermocycler (Bio-Rad, Hercules, CA, USA) with the following conditions: initial denaturation at 94 °C for 5 min, followed by 30 cycles of denaturation at 94 °C for 30 s, annealing at 50/55 °C (tRNAleu-COII/microsatellites) for 30 s, and elongation at 72 °C for 1 min with a final elongation at 72 °C for 10 min. All PCR products were examined on 8% polyacrylamide gels stained with ethidium bromide and observed under an ultraviolet transilluminator Gel Doc™ XR+ (BioRad, Hercules, CA, USA).

To determine the genetic structure of samples, the Structure 2.3.4 program was used with a given number of clusters from 1 to 10. The number of intended groups (K) was calculated in a Structure Harvester. The analysis was performed using the Admixture model with a burn-in period and MCMC equal to 10,000 and 100,000 repetitions, respectively. The results of the analysis were processed in CLUMPP 1.1.2 using the FullSearch algorithm. Genetic differentiation between populations was computed using unbiased estimates of FST values with GENEPOP.

## 3. Results

### 3.1. Assessment of Morphometric Parameters

The results of the morphometric analysis of the bees are presented in Table 2 and Table 3. It was established that, according to the length of tergite 3 and the length of the proboscis, all colonies of all subspecies and stocks corresponded to standard values. According to the cubital index, colonies of Prioksky bees corresponded not to the standard values of the breeding stock but to the subspecies *A. m. caucasica*. The cubital index for samples of *A. m. mellifera* was also below the standard value [39,40]. The cubital index of the Burzyan sample (AmmB), despite the fact that it was lower than standard values, corresponded to the CI for the Burzyan population of *A. m. mellifera* [41].

There are no standard values for other characters. We calculated the mean, minimum, and maximum values for our seven samples (Table 3). The highest value of the tarsal index (*TI*) was found in the Grey Caucasian Mountain bee and in the Prioksky bees. The samples also differed in the length of the wax mirror (*Lwm*)—the highest values were found in samples of *A. m. mellifera* (AmmB and AmmI). For samples Pr, Amcau, Amcarp, and Amcarn, this indicator was ≤1.40. In the Far Eastern sample, *Lwm* had an intermediate value (1.44). The same can be said about the distance between wax mirrors (*Lwmd*).

Thus, morphometric analysis showed that Prioksky bees corresponded to the subspecies *A. m. caucasica*. The Far Eastern bee had intermediate values between the samples of *A. m. mellifera* and subspecies from lineage C or O. Samples of *A. m. mellifera* from the Republic of Bashkortostan (AmmB and AmmI) did not correspond to standard values according to the cubital index. In particular, the AmmI sample, according to *CI*, corresponded to the subspecies *A. m. caucasica*. Perhaps this population underwent hybridization. In the sample from the Burzyansky district, *CI* was 57%, corresponding to the standard of the Burzyansky population of *A. m. mellifera*.

### 3.2. Assessment of the Genetic Structure of the Apis mellifera Samples

From the evolutionary lineage M on the maternal line came samples from the Iglinsky and Burzyansky districts of Bashkortostan, as well as one colony from the Krasnodar Krai, positioned as *A. m. carnica*. All other samples belonged to the C/O evolutionary lineages. Bees of the Prioksky breeding stock also descended on the maternal line from lineage C or O (with the markers used, we could not differentiate *A. m. caucasica* from lineage C subspecies).

The analysis of the polymorphism of microsatellite loci also showed that Prioksky bees, along with bees from the Krasnodar Krai and the Republic of Adygea, belonged to subspecies from the C/O lineages (Table 4, Appendix A). Samples from the Burzyansky and Iglinsky districts were confirmed to belong to *A. m. mellifera*. The level of introgression of the gene pool of the evolutionary lineages C/O in the AmmI sample was 18.4%, while in the sample AmmB it was 8%. All colonies from the Far Eastern sample originated on the maternal line from lineage C or O. The level of hybridization in this sample was 73.7%. Consequently, the gene pool of this sample was formed by subspecies from the evolutionary lineages M, C, and O, which is confirmed by the history of the settlement of this territory by bees of different subspecies.

We calculated pairwise Fst between the studied samples (Table 5). The greatest divergence was observed between samples of AmmI (dark forest bee *A. m. mellifera*, Iglinsky district) and Amcau (*A. m. caucasica*, Krasnopolyansk experimental station), i.e., 0.5736. The smallest divergence was between the samples AmmI and AmmB (*A. m. mellifera* from the Burzyansky district), i.e., 0.0133.

Phylogenetic analysis was also performed based on a polymorphism analysis of microsatellite loci using genetic distance estimation, according to Nei (1983) [42].

Thus, both samples from the Republic of Bashkortostan corresponded to the declared subspecies *A. m. mellifera* based both on the results of the analysis of the mitochondrial marker *tRNAleu-COII* and on the analysis of microsatellite markers. However, in the AmmI sample, the level of introgression of the C/O gene pool was 18.4%. The set of microsatellite loci we used was not capable of differentiating subspecies from the evolutionary lineages C (samples of *A. m. carnica* and *A. m. carpatica*) and O (*A. m. caucasica*). However, a phylogenetic analysis showed (Figure 1) that these samples formed different clusters: samples of *A. m. carnica* and *A. m. carpatica* were included in one cluster; samples of *A. m. caucasica* and Prioksky bees formed another cluster. The Far Eastern sample occupied an intermediate position between the samples of *A. m. mellifera* and *A. m. caucasica*.

### 3.3. Evaluation of Cryopreserved Sperm

In order to evaluate the effectiveness of the cryopreservation method used, some of the cryopreserved sperm was thawed after four days. A comparative assessment of the fresh (*N* = 100) and frozen–thawed (*N* = 27) sperm of drones of the Prioksky bees was performed (Table 6). The viability of frozen–thawed sperm (membrane integrity) in the test samples decreased by 20.3% compared to that of fresh sperm. Previously, we showed that honey exhibits cryoprotectant properties, maintaining sperm viability at a level of 37.2 ± 0.5%, and provides a sufficiently high protection of the vital resource of the sperm of honey bee drones in combination with 10% DMSO (79.6 ± 1.2%). Of the six queen bees artificially inseminated with frozen–thawed sperm based on a 10% honey extender, two yielded offspring of 96.5–99.1% worker bees [17].

The overall motility of frozen–thawed sperm was reduced 25-fold compared to that of fresh sperm. In order to determine the fertilizing ability of frozen–thawed sperm, we carried out artificial insemination of 10 virgin queens. During the assessment of the physiological parameters of AI queens, the presence of sperm in the spermatheca of four out of ten queens was revealed (Table 7).

An important physiological parameter of the queen bee, which determines its reproductive abilities, is the concentration of sperm in the seminal receptacle [43]. Previously [9,44], a positive correlation was identified between the concentration of sperm in the spermatheca of the queen and the number of fertilised eggs laid by the queen (r = 0.54 and r = 0.91). The concentration of sperm in the spermatheca of artificially inseminated queens was reported to vary from 1.8 million/μL [45] to 6 million/μL [46], and in naturally mated queens from 4 to 7 million/μL [43]. At the same time, sperm in the spermatheca of natural mating queens can be stored for several years. In our case, the concentration of sperm in the spermatheca of AI queens varied from 0.22 to 4.4 million/μL.

Thus, the reproductive period of the queen bee will depend on the number of sperm that entered its seminal receptacle. It is obvious that queen bees 1 and 3 will have a longer reproductive period (oviposition period) than queens 2 and 4. All four queen bees will produce more than 50% worker bee offspring. Queens Nos. 8–10 had no sperm in the spermatheca or paired oviducts due to insufficient sperm in the insemination dose. Queens Nos. 5–7 had no sperm in the spermatheca, but sperm were present in the paired oviducts. This appears to be related to the way sperm are stored after insemination [37].

## 4. Discussion

In this study, we provide an example of creating a biobank of the sperm of honeybee drones of certain subspecies. For subspecies identification, we used morphometric and genetic data. Genetic analysis showed that Prioksky bees, along with bees from the Krasnodar Krai and Adygea, belonged to subspecies from the C/O lineage. At the same time, phylogenetic analysis showed that the Prioksky bees formed one cluster with a sample of *A. m. caucasica* from the Krasnopolyansk experimental station. The Prioksky breeding stock was bred at the Federal Beekeeping Research Centre on the basis of crossing *A. m. mellifera* and *A. m. caucasica* [41,47]. The results of a genetic analysis showed that the Prioksky sample was more consistent with the subspecies *A. m. caucasica* and the level of the gene pool of *A. m. mellifera* in it was only 1.4%. Samples of *A. m. carnica* and *A. m. carpatica* also formed one cluster. Samples AmmB and AmmI were confirmed to belong to *A. m. mellifera*. The sample from the Far East was of hybrid origin. The Far Eastern breeding stock was created by crossing *A. m. mellifera*, *A. m. carpatica*, *A. m. caucasica*, *A. m. remipes*, and *A. m. ligustica* [39]. Thus, the genetic analysis confirmed the hybrid origin of this breeding stock.

Morphometric analysis also showed that Prioksky bees corresponded to the subspecies *A. m. caucasica*. The Far Eastern bees had intermediate values between the samples of *A. m. mellifera* and the samples *A. m. carnica*, *A. m. caucasica*, and *A. m. carpatica*. Samples of *A. m. mellifera* from the Republic of Bashkortostan did not correspond to standard values according to the cubital index. In particular, the AmmI sample, according to the cubital index, corresponded to the subspecies *A. m. caucasica*. Perhaps this population has undergone hybridization. According to the genetic analysis of the AmmI sample, the level of introgression of the C/O gene pool was 18.4%, which confirms hybridization. In the AmmB sample, the cubital index was 57%, which corresponds to the standard of the Burzyansky population of *A. m. mellifera*. The level of hybridization in this sample was only 8%. Previously, Oleksa and Tofilski (2015) [48] showed that morphometric and genetic methods yielded almost the same result—more than 90% of colonies were classified as one subspecies. These authors used 17 microsatellite loci and the mtDNA *COI-COII* locus as a genetic method, and as a morphometric method, they employed the geometric morphometry of the venation of the fore wing. In our study, the results of genetic and morphometric analyses were also consistent with each other.

An analysis of the viability and motility of cryopreserved sperm showed that the use of honey makes it possible to achieve significant progress in the preservation of reproductive gametes during low-temperature freezing. The percentage of viability of frozen-thawed sperm of our method with honey was consistent with the results of the method with egg yolk (69.75 ± 2.32%) [8] and royal jelly (68.9 ± 3.88%) [10]. The average number of spermatozoa in the spermatheca of queen bees using our method with honey (1.98 ± 0.9 millions/µL) slightly exceeded the result of the authors using the method with egg yolk [9] (1.6 × 10^6^ ± 4.7 × 10^4^) and significantly exceeded the result of authors [5] (134 ± 59 thousands spermatozoa) using the egg yolk and 25% DMSO method.

The main component of honey is carbohydrates (fructose, glucose, sucrose, maltose), dissolved in a small amount of water, as well as vitamins B1, B2, B6, E, K, C, carotene, and folic acid in small quantities. Honey has proven antimicrobial (antibacterial, antimycotic, antimycobacterial) properties, the interest in which has recently been growing [49]. The addition of honey to an extender for cryopreservation significantly improves sperm motility after thawing and the integrity of membranes and acrosomes and also reduces the number of abnormalities in sperm morphology in horses [50], bulls [51], buffalo [52], goats [53], sheep [54], rats and mice [55,56], fish [57,58], and humans [59].

When using honey as a cryoprotectant, it is necessary to take into account the fact that the physicochemical composition of honey also depends on its botanical origin. This circumstance may explain the different results obtained by other researchers when using honey as the main extender. For example, Malik et al. (2017) [51], experimenting with the replacement of glycerol with honey at a concentration of 8%, revealed a significant increase in abnormalities in sperm morphology (8.35 ± 0.16%), while the viability (82.19 ± 1.41%) and motility (76.63 ± 3.21%) did not differ significantly from those of fresh sperm. Shikh Maidina et al. (2018) and Fanni et al. (2018) [53,57], on the contrary, did not find major changes in sperm morphology but noted frequent changes in the head and tail of the flagellum. Other authors [54] reported a lower percentage of dead and abnormal sperm when using honey.

## 5. Conclusions

Using morphometric and genetic methods, we selected the sperm of the drone bees of different subspecies. A honey solution was used as a cryopreservative. It was shown that the main indicators of frozen–thawed sperm decreased compared to those of fresh sperm. However, the use of honey in sperm cryopreservation has great potential. With the help of cryopreservation, it will be possible to preserve endangered subspecies of *Apis mellifera* L.

## Figures and Tables

**Figure 1 animals-13-03684-f001:**
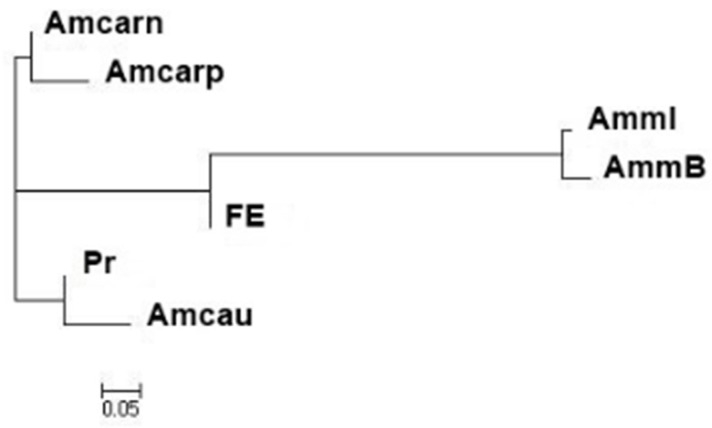
Dendrogram of the genetic relatedness of the studied samples of honey bees (Nei, 1983) [42].

**Table 1 animals-13-03684-t001:** Characteristics of the studied samples.

Sampling Region	Sample Name	*N*	Subspecies
The Republic of Bashkortostan, Iglinsky district	AmmI	10	*A. m. mellifera*
The Republic of Adygea, Maykop	Amcarn	8	*A. m. carnica*
The Republic of Adygea, Maykop	Amcarp	10	*A. m. carpatica*
Far East, Primorsky Krai (Kondratenovka and Tichoreshnoe villages)	FE	10	*Hybrid of A. m. mellifera*, *A. m. carpatica*, *and A. m. caucasica*
The Republic of Bashkortostan, Burzyansky district, Shulgan-Tash Nature Reserve	AmmB	9	*A. m. mellifera*
Ryazan region, Rybnoye, The Federal Beekeeping Research Centre apiary	Pr	10	*Prioksky bees*, *hybrid of A. m. mellifera and A. m. caucasica*
Krasnodar Krai, Adler, Krasnopolyansk experimental beekeeping station	Amcau	19	*A. m. caucasica*

**Table 2 animals-13-03684-t002:** Morphometric characteristics of the study and reference samples.

Sample	*N*	*CI*, %	*Lx*, mm	*Wt3*, mm
Pr	10	50.5	6.88	4.84
**Standard values for Prioksky bees**		**55–60**	**6.6–6.9**	**4.6–5.0**
AmmI	10	52.9	6.17	4.92
AmmB	9	57.0	6.21	4.91
**Standard values for *A. m. mellifera***		**60–65**	**6.0–6.4**	**4.8–5.2**
Amcarp	10	41.9	6.63	4.81
**Standard values for *A. m. carpatica***		**33–43**	**6.3–7.0**	**4.4–5.1**
Amcarn	8	39.4	6.71	4.90
**Standard values for *A. m. carnica***		**<40.0**	**6.4–6.8**	**4.7–5.1**
FE	10	43.9	6.53	4.97
**Standard values for Far Eastern bees**		**28–60**	**6.1–6.8**	**4.6–5.4**
Amcau	15	52.4	7.01	4.80
**Standard values for *A. m. caucasica***		**50–55**	**6.7–7.2**	**4.4–5.0**
** *A. m. mellifera ** **		**61.4**	**6115**	**-**
** *A. m. carnica ** **		**51.2**	**6458**	**-**
** *A. m. caucasica ** **		**54.7**	**6976**	**-**

*—values from source [40].

**Table 3 animals-13-03684-t003:** Morphometric characteristics of the study samples.

Sample	*N*	*F_L_*, mm	*F_W_*, mm	*TI*, %	*Ls3*, mm	*Lwm*, mm	*Lwm*, mm	*Lwmd*, mm	*Lt3*, mm
Pr	10	9.43	3.16	56.5	2.86	1.39	2.48	0.29	2.29
AmmI	10	9.34	3.09	54.6	2.92	1.50	2.55	0.22	2.36
AmmB	9	9.37	3.09	55.6	2.91	1.50	2.53	0.21	2.33
Amcarp	10	9.30	3.12	55.7	2.82	1.39	2.46	0.29	2.25
Amcarn	8	9.36	3.14	55.1	2.85	1.40	2.50	0.29	2.27
FE	10	9.37	3.16	55.5	2.93	1.44	2.53	0.27	2.34
Amcau	15	9.40	3.13	56.8	2.85	1.39	2.47	0.31	2.27
**Mean**		**9.37**	**3.13**	**55.7**	**2.88**	**1.43**	**2.50**	**0.27**	**2.30**
**Min**		**9.30**	**3.09**	**54.6**	**2.82**	**1.39**	**2.46**	**0.21**	**2.25**
**Max**		**9.43**	**3.16**	**56.8**	**2.93**	**1.50**	**2.55**	**0.31**	**2.36**

**Table 4 animals-13-03684-t004:** Results of the genetic analysis of the studied samples of *Apis mellifera*.

Sample	Allelic Variant of *tRNAleu-COII*	Gene Pool of C/O	Gene Pool of M
M lineage	98 PQQ, 38 PQQQ	0.016	0.984
C/O lineage	120 Q	0.993	0.007
AmmI	10 PQQ	0.184	0.816
Amcarn	7 Q, 1 PQQ	0.946	0.054
Amcarp	10 Q	0.967	0.033
FE	10 Q	0.737	0.263
AmmB	9 PQQ	0.080	0.920
Pr	10 Q	0.986	0.014
Amcau	19 Q	0.988	0.012

**Table 5 animals-13-03684-t005:** Pairwise Fst between samples (* = *p* < 0.05; NS = not significant).

	AmmI	Amcarn	Amcarp	FE	AmmB	Pr	Amcau
**AmmI**	0.0000	*	*	*	NS	*	*
**Amcarn**	0.4174	0.0000	NS	*	*	NS	*
**Amcarp**	0.4602	0.0389	0.0000	*	*	*	*
**FE**	0.2546	0.1140	0.1959	0.0000	*	*	*
**AmmB**	0.0133	0.4682	0.5177	0.3030	0.0000	*	*
**Pr**	0.4414	0.0944	0.1606	0.2014	0.4677	0.0000	*
**Amcau**	0.5736	0.2466	0.3267	0.4059	0.5843	0.1311	0.0000

**Table 6 animals-13-03684-t006:** Main indicators of the sperm quality of Prioksky drones before and after four days of cryopreservation.

Indicator	Frozen–Thawed Sperm (*N* = 27)			Fresh Sperm (*N* = 100)		
	M ± m (Min–Max)	σ	Cv, %	M ± m (Min–Max)	σ	Cv, %
Motility, %	2.2 ± 0.6 (0–11.5)	3.1	141.04	55.0 ± 2.6 (0–99.8)	26.5	48.3
Viability, %	64.0 ± 1.8 (41.5–83.7)	9.6	14.87	84.3 ± 1.2 (40–99.9)	12.2	14.5

**Table 7 animals-13-03684-t007:** Physiological parameters of Prioksky queen bees artificially inseminated with frozen–thawed sperm.

No. of AI Queens	Concentration of Sperm in the Spermatic Receptacle, millions/μL	Presence of Sperm in Paired Oviducts
1	2.4 ± 0.25 (2.2–2.7)	absent
2	0.9 ± 0.3 (0.6–1.2)	absent
3	4.4 ± 0.3 (4.1–4.7)	absent
4	0.22 ± 0.02 (0.2–0.25)	sperm traces
5	0	large amount of sperm
6	0	large amount of sperm
7	0	large amount of sperm
8	0	absent
9	0	absent
10	0	absent

## Data Availability

The data presented in this study are available in the Appendix A.

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
