# Peer review of "Creation of a Biobank of the Sperm of the Honey Bee Drones of Different Subspecies of Apis mellifera L."

_animals, 2023, doi:10.3390/ani13233684_

Round 1

Reviewer 1 Report

Comments and Suggestions for Authors

This study provides an example of the creation of a biobank of drone sperm of specific subspecies of Apis mellifera L. The measurement of the sperm concentration in the spermatheca of artificially  inseminated queens showed that it varied from 0.22 to 4.4 million/μL. The results showed that the use of honey in sperm cryopreservation has great potential.

Here I only suggested some minor suggestions.

1.    Materials and Methods, samples used for morphometric analysis are not clear. In sampling part, workers and drones are collected. But which one was used in morphometric analysis did not mentioned.

2.    For creation of a biobank of the sperm, the viability of frozen–thawed sperm is the most important thing. And the method to improve the viability of frozen–thawed sperm and the standard of the viability of frozen–thawed sperm should meet did not studied in this research. This is critical in creating a biobank of the sperm.

Author Response

Dear Reviewer,

Thank you very much for your detailed review of our work. We have corrected the article  according to your comments.

The table below contains the responses to your comments.

 Materials and Methods, samples used for morphometric analysis are not clear. In sampling part, workers and drones are collected. But which one was used in morphometric analysis did not mentioned.

We moved part of the text (from line 179) indicating which individuals were selected for analysis to the beginning of the text and added a link to an article describing the characteristics under study ([27] https://doi.org/10.3896/IBRA.1.52.4.05)

For creation of a biobank of the sperm, the viability of frozen–thawed sperm is the most important thing. And the method to improve the viability of frozen–thawed sperm and the standard of the viability of frozen–thawed sperm should meet did not studied in this research. This is critical in creating a biobank of the sperm.

In this study, we did not set the goal of increasing the viability of frozen-thawed sperm. The goal of this study was to create a biobank of bee sperm of different subspecies using a method that we had previously developed. But we will certainly explore this issue in our future work. We were unable to find relevant information about honey bee drone sperm standards from other authors. However, we made a small comparison of our data with the results of other authors in the text of the article: lines 381-388 in the Discussion.

Reviewer 2 Report

Comments and Suggestions for Authors

In the manuscript "Creation of a biobank of the sperm of the honey bee drones of different subspecies of Apis mellifera L. using a honey extender", Gulov et al present a study in which they used morphometric and genetic methods to identify subspecies of honey bees. Moreover, the authors tested honey dilution as a cryopreservative for honey bee drone sperm storage. Overall, the rationale for their study is well defined. However, I have some concerns as outlined below:

1. The authors described several previously used sperm cryopreservative (eg. 25% DMSO, egg yolk, C46 nutrient medium), however, it is not clear to me that how much better does their new method of sperm preservation perform comparing to previous methods. It would be easier to follow if the authors can summarize the sperm viability, mobility and fertility reported for previous methods, together with the performance of their new method, maybe in a summary table.

2. In the Methods, the authors describe their cryopreservation solution is a mixed solution including several previously used cryoprotectant “10% honey (50 mL), lactose (10 mg), sucrose (10 mg), egg yolk (2.5 mL), and DMSO (5 mL)”. Therefore, it is more accurate to state that their new method is adding honey into the cryopreservation solution, instead of simply stating honey diluent as sperm cryopreservation.

3. In Table 7, queen 1-4 show 0.22 to 4.4 million/μL sperm concentration in spermatheca. But queen 5-10 show 0 sperm in spermatheca. 5-7 show sperm in paired oviducts, and 8-10 show nothing in either spermatheca or paired oviducts. The authors should explain what does this observation mean.

Author Response

Dear Reviewer,

Thank you very much for your detailed review of our work. We have corrected the article  according to your comments.

The table below contains the responses to your comments.

The authors described several previously used sperm cryopreservative (eg. 25% DMSO, egg yolk, C46 nutrient medium), however, it is not clear to me that how much better does their new method of sperm preservation perform comparing to previous methods. It would be easier to follow if the authors can summarize the sperm viability, mobility and fertility reported for previous methods, together with the performance of their new method, maybe in a summary table.

We did not evaluate the reproductive parameters of queen bees (% of the brood of workers) inseminated with frozen-thawed sperm. Of course, a summary table of data from other authors would allow a more objective assessment of the effectiveness of our cryopreservation method. We made a similar table, but since the measured indicators differ from different authors, it was impossible to compare them. We believe that in this case our comparative assessment will look incorrect. However, we made a small comparison of our data with the results of other authors in the text of the article: lines 381-388 in the Discussion.

In the Methods, the authors describe their cryopreservation solution is a mixed solution including several previously used cryoprotectant “10% honey (50 mL), lactose (10 mg), sucrose (10 mg), egg yolk (2.5 mL), and DMSO (5 mL)”. Therefore, it is more accurate to state that their new method is adding honey into the cryopreservation solution, instead of simply stating honey diluent as sperm cryopreservation.

We have made the necessary changes - "honey extender" was replaced with "cryoprotectant solution with honey" :

- in the title of the article;

- in Abstract line 31;

- in Introduction line 76.

In Table 7, queen 1-4 show 0.22 to 4.4 million/μL sperm concentration in spermatheca. But queen 5-10 show 0 sperm in spermatheca. 5-7 show sperm in paired oviducts, and 8-10 show nothing in either spermatheca or paired oviducts. The authors should explain what does this observation mean.

We explained what this observation means: « Queen 8-10 shows 0 sperm in spermatheca or in the paired oviducts due to insufficient sperm in the insemination dose. Queen 5-7 show 0 sperm in spermatheca and show sperm in paired oviducts are apparently associated with the way they are kept after insemination [37].». Lines 342-345.
